# Comparison of Bone Mineral Density and Trabecular Bone Score in Patients with and without Vertebral Fractures and Differentiated Thyroid Cancer with Long-Term Serum Thyrotrophin-Suppressed Therapy

**DOI:** 10.3390/diagnostics14090868

**Published:** 2024-04-23

**Authors:** Federico Hawkins Carranza, Cristina Martin-Arriscado Arroba, María Begoña López Alvarez, Soledad Librizzi, Guillermo Martínez Díaz Guerra

**Affiliations:** 1Research Institute i+12, University Hospital 12 de Octubre, Faculty of Medicine, University Complutense, 28041 Madrid, Spain; guillermo.martinez@salud.madrid.org; 2Research Institute i+12, University Hospital 12 de Octubre, 28041 Madrid, Spain; cmartinarriscado.imas12@h12o.es; 3Association Investigation of Osteoporosis and Endocrine Diseases, 28020 Madrid, Spain; mariabegona.lopezal@salud.madrid.org; 4Service of Endocrinology, University Hospital 12 de Octubre, 28041 Madrid, Spain; solelibrizzi@hotmail.com

**Keywords:** differentiated thyroid cancer, TSH suppression therapy, bone mineral density, trabecular bone score, levothyroid

## Abstract

Introduction: The study of BMD provides only partial information on bone health in patients undergoing TSH suppression therapy due to differentiated thyroid cancer (DTC). The trabecular bone score (TBS), a new parameter assessing bone microarchitecture, is proposed for studying bone in this context. This study aimed to analyze their long-term use in patients with DTC. Methods: Bone mineral density (BMD) was measured by dual X-ray densitometry (DXA) and TBS was assessed with iNsigth software (version 2.0, MediImaps, France) in 145 postmenopausal patients with DTC. Vertebral fractures (VFs) were identified using a semi-quantitative X-ray method. Results: The BMD at the end of this study did not differ from the initial measurement. However, the TBS decreased from 1.35 ± 0.1 to 1.27 ± 0.1 (*p* = 0.002). Increased levels of PTH, osteocalcin, and bone alkaline phosphatase (BAP) were observed, suggesting enhanced bone remodeling. There was an increase in the prevalence of osteoporosis and osteopenia (40.6% and 16.5% to 46.6% and 18.6%, respectively). The proportion of patients with partially degraded and totally degraded TBS increased from 31% and 15.1% to 48.9% and 24.8% by the end of this study. Among the 30 patients with VFs, there were no significant differences in age, body mass index (BMI), calcium intake, alcohol consumption, smoking, radioiodine, therapy, or thyroid parameters compared to those without VFs. The odds ratio for VFs increased with osteopenia (OR 2.63). Combining TBS with BMD did not improve discrimination. Conclusions: The TBS decreased while the BMD remained unchanged. The percentage of patients with osteoporosis and osteopenia, whether partially degraded or totally degraded, increased by the end of this study. The predominant discordance was found in partially degraded microarchitectures, with a higher proportion of osteopenic patients compared to those with normal or osteoporotic bone density. The AUC of the combination of TBS and BMD did not enhance discrimination. TBS, radioactive iodine therapy, and sedentary lifestyle emerged as the main distinguishing factors for DTC patients with VFs.

## 1. Introduction

Thyroid carcinoma ranks as the most prevalent endocrine malignancy, with its increased prevalence attributed to the growing utilization of diagnostic and imaging techniques [1]. Differentiated thyroid carcinoma (DTC) are a majority in this group, in which TSH-suppressive therapy employing levothyroxine (LT4) is recommended to counteract the stimulatory effects of TSH [2]. The current guidelines advocate for thyroidectomy followed by radioactive iodine therapy and long-term TSH suppression with LT4 for DTC management. It has been noted that postmenopausal women with DTC may face a heightened risk of bone loss and fractures due to the suppressive impact of TSH [3]. However, such risks have not been definitively established for men and premenopausal women. The mechanism underpinning bone loss remains poorly understood, but it is posited that suppressed TSH levels could impede bone formation while promoting osteoclast differentiation, leading to dysregulated bone turnover and microarchitectural abnormalities [4]. Additionally, a high prevalence of VFs has been reported and is now considered the most common fracture type associated with this condition [4].

While dual-energy X-ray absorptiometry (DXA) remains the gold technique for measuring bone mineral density (BMD) and diagnosing bone loss, it is crucial to acknowledge a significant limitation: the majority of fractures occur in osteopenic women or even in individuals with normal BMD [5]. Additionally, it is widely recognized that BMD measurements may be elevated in the presence of osteoarthritis, vascular and joint calcification, and lumbar scoliosis [6]. As a result, relying solely on BMD measurements may not accurately estimate the overall impact of a disease or medication on bone health.

There is an ongoing effort to determine whether evaluating bone quality can enhance the identification of high-risk patients for fractures, in addition to DXA. Various techniques have been proposed for analyzing bone quality, including trabecular bone score (TBS), semiquantitative radiology, QCT, MRI, microindentation, and SHAFT 3D. The TBS offers the advantage of being non-invasive and effective. TBS involves a gray-scale textural analysis of anteroposterior Lumbar Spine DXA scans, typically obtained for BMD measurement. It is sensitive to changes over time, including response to treatments and progression of bone disease across a wide range of secondary osteoporosis conditions. The TBS has demonstrated its ability to capture morphometric properties of vertebral bone by pixel-based gray variations, thereby providing information beyond what DXA alone can offer. As a BMD-independent predictor of skeletal strength and fracture risk, TBS can complement the prediction of future osteoporotic fractures [7].

This study aimed to analyze changes in TBS, BMD, and vertebral fractures (VFs) following a long-term follow-up of postmenopausal patients who underwent total thyroidectomy due to DTC (papillary and follicular types) and were subjected to L-thyroxine (LT4)-suppressive therapy.

## 2. Patients and Methods

### 2.1. Study Population

We conducted a study involving postmenopausal patients who had undergone total thyroidectomy due to DTC (papillary and follicular types) and had undergone DXA three months after surgery. These patients were subjected to a mean follow-up of 10 years with TSH suppression using LT4 at our site. They were regularly monitored at our Thyroid Unit with a follow-up every 6–12 months. Patients were eligible for inclusion if they reached one-year post-menopause and had maintained TSH suppression (<0.5 µU/mL) during 80% of their visits to our site. Informed consent was obtained from all patients. Those who had taken medications affecting bone metabolism (calcium or vitamin D, antiresorptive, hormone replacement therapy) or had other conditions potentially impacting bone metabolism (hyper and hypoparathyroidism, hyper and hypothyroidism, rheumatoid arthritis, asthma, Paget disease) were excluded from the study, as were those with abnormal kidney or liver function. Additionally, participants completed standard questionnaires regarding calcium intake and physical activity. The study protocol was approved by the Ethical Committee of the University Hospital 12 de Octubre, Madrid, Spain.

### 2.2. Laboratory Determinations

Fasting serum samples were obtained at the baseline study after 3 months of thyroidectomy and suppressive TSH therapy and at the end study, between 8 and 9 AM. These samples were immediately frozen at −70 °C until the assays were conducted. The following parameters were measured using automated standard laboratory methods (Modular P800; Roche Diagnostic, Budapest, Hungary): Serum calcium corrected for albumin binding (normal range 8.6–10.2 mg/dL), phosphate (normal range 2.5–4.5 mg/dL), and creatinine (normal range 0.70–1.2 mg/dL). Serum thyrotrophin (TSH) was measured by chemiluminescence (Architect, Lake Forest, IL, USA) (normal range: 0.5–4.5 µIU/mL), free T4 (FT4) by electrochemiluminescence (E, CLIA, London, UK) (functional sensitivity < 0.01 μg/mL) (normal range: 0.92.3 ng/dL), and thyroglobulin using Immulite 2000 (functional sensitivity 0.9 ng/mL) (normal range 0.5–5.0 ng/mL). Serum intact parathyroid hormone (PTH) levels were determined using chemiluminescent immunoassays (IMMULITE 2000, DPC, Berlin, Germany) (normal range: 7–57 pg/mL). Serum 25–hydroxyvitamin D3 (25OHD) was measured by enzyme immunoassay (automated IDS EIA, Biotechne, Barcelona, Spain). Vitamin D deficiency was considered for values below 20 ng/mL. Osteocalcin (OC) (N-MID Osteocalcin, Roche Diagnostics) was determined with electrochemiluminescence (normal range: 8–48 ng/mL). Bone alkaline phosphatase (BAP) (IDS EIA) (normal range:15–40 U/L) and β-cross-laps (β-CTX) were determined using ELECSYS 1010, Roche electrochemiluminescence (normal range: 0.2–0.704 ng/mL) (Basil, Swiss).

BMD was assessed by DXA (densitometer QDR 4500, Hologic, Waltham, MA, USA). Measurements were taken at various skeletal sites, including the lumbar spine (L-BMD), femoral neck (FN-BMD), total hip (TH-BMD), and distal third of the radius (1/3DR-BMD). The coefficient of variation was 0.90% for LS-BMD and 0.70% for FN-BMD. The least significant change was 1.8% for L-BMD and 2.9% for TH-BMD. BMD values were expressed as absolute values (g/cm^2^) and as standard deviations (SDs) from the expected adult BMD represented as T scores. Reference databases specific to the Spanish population were derived from a multicenter densitometric study involving 1305 healthy women, aged 20–80 years [8]. Patients underwent identical bone densitometry scans initially and subsequently, utilizing the same DXA system. The diagnoses were made according to the WHO criteria (based on T scores); osteoporosis was diagnosed if an individual’s BMD was equal to or less than −2.5 standard deviations below the mean normal BMD value of healthy young adults; osteopenia was diagnosed if the T scores fell between −1≥ and >−2.5; and normal bone density was indicated by a T score greater than −1 [9].

Trabecular bone score (TBS) measurements were conducted using the TBS iNsight 2.0 software (Med-Imaps, Pessac, France), applied to the L-BMD DXA exams. Lumbar TBS was determined as the mean value of individual measurements for vertebrae L1–L4. Patients were categorized into three groups based on reference values: TBS ≥ 1.350 was considered normal; TBS between 1.350 and 1.200 indicated partially degraded microarchitecture; and TBS ≤ 1.200 indicated totally degraded microarchitecture [10]. The coefficient of variation of TBS calculated from 3 repeated measurements in 15 women was 0.8%.

During the study, lateral and anteroposterior radiographs of the thoracic and lumbar spine were obtained in the upright position for all subjects. Vertebral fractures were identified on X-ray films using the method described by Genant et al., as assessed by a trained radiologist [11]. 

### 2.3. Statistical Analysis

Quantitative variables were expressed as mean and standard deviation or median and interquartile range. The normal distribution pattern was confirmed by the Kolmogorov–Smirnov test. Qualitative variables were described as absolute and relative percentages. Contingency tables and Chi-square or Fisher Exact test were used to compare categorical parameters. McNemar’s test or Simmetry’s test compared the change in the distribution of classification of patients with thyroid cancer according to DXA and TBS scores. The non-parametric U-Mann–Whitne–Wilcoxon test or Student’s *t*-test was used for the cross-sectional study and Student’s *t*-test for the longitudinal study. Pearson correlation was performed to assess correlations between clinical parameters and DXA and TBS parameters. 

Linear regression was performed to evaluate the dependence and influence between variables with the TBS and BMD for the lumbar spine, expressed by the coefficient and 95% confidence interval. Also, a logistic regression was carried out to study the association between the possible risk factors and the final fractures, expressing the results in odds ratio (OR) together with the 95% confidence interval (95%CI). The statistical study was completed with multivariate analysis, considering both those risk factors with a significant result in the univariate analysis and those that had a certain relevance within the objectives of this study. The use of stepwise selection was employed to highlight the most relevant factors, identify those significant sets among the possible variables, and avoid confounding the model due to possible relationships between them since the method determines the significance of the effect of a variable in the presence of those already in the model. The accuracy, along with its 95% confidence interval, was estimated using the ROC curve. 

All analyses were conducted using Stata InterCooled for Windows version 16 (StataCorp. 2019. Stata Statistical Software: Release 16. College Station, TX, USA: StataCorp LLC) and SAS for Windows version 9.4 (SAS Institute Inc., Cary, NC, USA) and *p* values < 0.05 were considered to indicate statistical significances in all statistical procedures.

## 3. Results

Table 1 presents the baseline and final data of 145 postmenopausal Caucasian patients with DTC who met the selection criteria. Serum calcium, phosphate, and creatinine were in the normal range at baseline and at the end of the study. The mean age of patients at surgery was 51 years. The mean BMI was slightly higher at the end of the study, after a mean follow-up of twelve years (28 ± 45 kg/m^2^ vs. 27 ± 0.6 kg/m^2^). Mean doses of LT4 were lower, and there was a significant increase in serum TSH levels at the end of the study (0.89 ± 0.30 µU/mL vs. 0.23 ± 0.41 µU/mL). The serum levels of free T4 did not change. Additionally, a significant increase in serum PT and in bone markers (serum osteocalcin, bone alkaline phosphatase) was observed at the end of the study. Beta-CrossLaps, which were only measured at the end study, were within the normal postmenopausal reference values. There was a non-significant tendency toward lower BMD values at the lumbar, hip, and radial sites. TBS values were significantly decreased at the end of the study, from 1.35 ± 0.14 to 1.27 ± 0.13. Initially, patients with DXA-diagnosed osteoporosis (*n* = 24) had lower TBS values (1.28 ± 0.11) compared to those with normal-osteopenia BMD values (*n* = 121, 1.36 ± 0.14, *p* = 0.029), both initially and at the end of the study (Figure 1). 

Patients were classified according to the DXA and TBS study. At baseline (Table 2), 42.7% had DXA scores within the normal range, compared to 40.6% and 16.5% in the osteopenia and osteoporosis range, respectively. At the end of the study (Table 3), 34.4% had normal DXA values, while 46.6% and 18.65% fell into the osteopenia (*n* = 69) and osteoporosis (*n* = 27) categories, respectively. Therefore, there was an increase in the number of patients with osteoporosis and osteopenia. At baseline, 31% of patients had partially degraded TBS values, while 15.1%, had totally degraded values, and 53.7% had normal scores. At the end of the study, the number of patients with partially degraded and totally degraded TBS saw a significant increase to 48.9% and 24.8%, respectively, while the number of patients with normal TBS scores saw a decrease to 26.2%. 

When comparing the differences in TBS and DXA scores changes (Table 4), we observed an increase in patients with degraded TBS (36 at the end of the study, compared to 22 at baseline), which was associated with osteopenia and osteoporosis. Additionally, there was a significant decrease in TBS values in this group (1.22 ± 0.07) compared to the group with normal-osteopenia BMD (1.29 ± 0.13, *p* = 0.019). Therefore, the predominant discordance rate of TBS with DXA was found in patients with a partially degraded microarchitecture, wherein the proportion of patients with osteopenia was higher than that of normal or osteoporotic patients (Figure 2). At the end of the study, we conducted a questionnaire regarding calcium intake. Interestingly, we found no differences in this parameter among the groups categorized as normal (561 ± 195 mg/d), osteopenia (572 ± 301 mg/d), and osteoporosis (588 ± 294 mg/d).

The association between TBS and lumbar DXA scores at baseline and at the end of the study was found to be significant when analyzed with Fisher´s exact test (<0.0001). Additionally, the symmetry between Table 2 and Table 3 was confirmed (*p* < 0.0001). Partial analysis revealed significant results when comparing the T score of Lumbar BMD with TBS values at the baseline study (Chi-Square 0.0057) and at the end of the study (<0.0001). Furthermore, the analysis of the same variables, comparing TBS values at baseline with TBS values at the end of the study, confirmed an association (Test of Symmetry <0.0001). However, this was not found with T scores of lumbar BMD, at baseline and at the end of the study (Test of Symmetry 0.2123).

When comparing the incidence of VFs found in 30 patients during the follow-up (Table 5), with patients without VFs (*n* = 115) there were no differences in age, BMI, smoking, calcium intake, or alcohol ingestion. Osteoporosis was diagnosed in 13 of the patients (11.3%) without VFs, and in 4 of 30 patients with VFs (13.3%). Thyroid parameters as well as bone markers were also similar. Three patients with DXA in the osteoporosis range and partially degraded TBS had VFs, while there were 17 in the densitometric osteopenic range (three normal, nine with partial, and five with totally degraded TBS). Among the 10 patients with normal DXA values, two had a totally degraded TBS (Table 6). 

There were no differences in VFs between patients with strong TSH suppression levels (*n* = 13, <0.1 µU/mL) versus those with moderate TSH suppression levels (*n* = 17, 0.1–0.5 µU/mL). When comparing TBS and DXA T scores at the end of the study, among the 30 patients with vertebral fractures, significant changes in the values were observed, as shown by Fisher’s Exact Test (*p* = 0.0009). At baseline, L-BMD was positively correlated with TBS (r = 0.38, *p* = < 0.001), while the latter variable was inversely correlated to BMI (r = −0.35, *p* = 0.0003). At the end of the study, an inverse correlation was found between L-BMD and BMI (r = 0.32, *p* = 0.0001), and a weak correlation with 25OHD vitamin D levels (r = −0.17, *p* = 0.45). As expected, TBS was also inversely related to BMI at the end of the study (r = −0.41, *p* = 0.0001). Bone markers (osteocalcin, beta-CrossLaps) were not correlated with any variable at baseline and the end of the study. There was no correlation between L-BMD and free T4 levels at baseline or the end of the study, while the TBS was inversely correlated with free T4 levels (r = −0.20, *p* = 0.040) at baseline but not at the end of the study. The AUC for the TBS plus BMD model was 0.5261 (95% CI 0.4423–0.6679) (Figure 3) and did not improve the discrimination of TBS over TBS alone. In the multivariate logistic regression analysis, using VFs as the dependent variable, the odds ratio for incidental fractures was 0.96 (95% CI 0.93–0.99, *p* = 0.037), since diagnostic to the end of the DXA study, with an OR of 1.08, (95% CI 1.02–1.14, *p* = 0.004), and for TSH levels an OR of 2.32 was found (95% CI 1.074–5.01, *p* = 0.032). The odds ratio for incident fractures was increased in patients having osteopenia (OR 2.63, 95% CI 1.11–6.24, *p* = 0.028). For BMD T-scores, TBS OR was significant at totally degraded (4.33, CI 1.079–17.388, *p* = 0.039) and partially degraded (4.06, 95% CI 1.2277–13.489, *p* = 0.022).

## 4. Discussion

In our study, we observed a slight non-significant decrease in BMD in postmenopausal women with DTC and TSH suppression after 12 years of follow-up. In contrast, TBSs were significantly decreased during this period. This finding is consistent with previous reports, indicating that significant BMD loss is typically observed after a prolonged duration of the disease [12]. Furthermore, we noted a great reduction in bone microarchitecture among patients who experienced incidental fractures over this extended period. Additionally, there were lower serum levels of 25OHD levels and an increase in serum PTH along with elevated serum levels of osteocalcin and BAP, which are markers of bone formation. However, serum levels of β-CTX, a marker of bone resorption, remained within normal range. This suggests a subtle imbalance over time in bone turnover, which may contribute to the observed trend toward lower BMD values at the end of the study.

There were no differences in dietary calcium intake and smoking among our patients between the baseline and the end of the study. The average calcium intake fell within the range of mean calcium intake (629 ± 290 mg/day) reported in a large Spanish population study [13]. Additionally, no significant difference in calcium intake was noted between fractured (477.50–750 mg/d) and non-fractured (250–750 mg/d) patients. Our patients exhibited low serum levels of 25OHD, both at baseline and at the end of the study. Hu et al. reported an inverse correlation between serum 25OHD related to thyroid cancer risk, with an OR of 1.42 (95 CI 1.17–1.73) in individuals with vitamin D deficiency compared to those with normal levels [14]. Furthermore, in a study involving 134 postmenopausal women with DTC, we found that 43% of patients had serum 25OHD deficiency. This deficiency was associated with an abnormal trabecular microarchitecture, compared to patients with normal serum 25OHD levels, suggesting a contribution of lower 25OHD levels to increased fracture risk [15].

In studies with a two-year follow-up period, it has been reported that TSH-suppressive therapy does not induce a significant difference in BMD compared to healthy controls [12,16,17]. However, in contrast, a meta-analysis revealed lower-spine BMD in 177 postmenopausal patients, with a duration of TSH suppression exceeding 5 years, compared with 69 patients with less than 5 years of treatment duration [18]. Additionally, it has been reported that the TBS in patients with DTC treated with long-term TSH suppression exhibits lower values, which can predict the relative fracture risk in these patients [19,20]. 

In a study comparing 43 patients with DTC with suppressed TSH levels to 20 individuals on LT4 replacement for less than 4 years, it was found that TBS did not deteriorate [21]. However, upon analysis of 31 postmenopausal patients, significantly lower degraded TBS values were observed compared to 32 premenopausal patients. Moon et al.’s data demonstrated that 4.2 years of TSH suppression in postmenopausal DTC patients led to a significant decrease in TBS, independent of changes in BMD [20]. In our study, moderate TSH suppression of TSH ranging from 0.1 to 0.5 µUI/mL was associated with lower TBS values [22]. These findings suggest that earlier damage to trabecular architecture may increase the susceptibility to bone fractures in this disease. Patients with a higher TBS may exhibit better trabecular architecture, thereby making them potentially more resistant to such events. 

We detected VFs during the follow-up in 20.6% of our patients with DTC. The data on fracture risk in patients with DTC undergoing TSH suppression therapy are limited. Our study was not specifically designed to evaluate incidence. Instead, we conducted a semiquantitative visual assessment of spinal X-rays. 

There have been two small cross-sectional studies involving postmenopausal women who underwent total thyroidectomy for DTC-studied VFs. In the first study by Fujiyama et al., no difference in incidental VFs was found between 12 patients with TSH suppression compared to 12 non-suppressed patients [23]. In another study by Heijckmann et al., a prevalence of 7% of VFs was reported in 59 patients, concluding that this rate was lower than the 12% in the European population [24]. On the other hand, in a larger study by Mazziotti et al., VFs were assessed using a quantitative morphometric approach in 178 postmenopausal women treated for DTC for at least one year. VFs were detected in 51 (28%), with older age, osteoporosis, TSH level < 0.1 mU/L, and duration of LT4 treatment as significantly associated. However, in this study, TSH was measured on a single occasion, and patients with VFs had a longer duration of treatment and lower BMD compared to those without [4]. 

In the NHANES 2005–2006 analysis, it was demonstrated that older women with higher levels of physical activity had a higher TBS [25]. Our patients with DTC and VFs engaged in lower daily exercise (which may result in less stimuli for bone remodeling), with no difference observed in age, smoking, alcohol consumption, or calcium intake. Recently, in a larger study involving 74.774 thyroidectomized patients with a mean follow-up period of 4.5 years, the group that had engaged in exercise showed a significant decrease in the risk of any fracture (VFs: HR 0.848) compared to those who lacked physical activity [26]. Therefore, the lack of physical activity may contribute to a lower TBS and increased fracture risk. Histology type, thyroglobulin levels, TNM stage, as well as radioactive iodine were not found to be significantly different. 

The discovery that patients with DTC exhibit a lower TBS, even in the presence of normal BMD, holds significant clinical implications. While it is generally assumed that patients with normal BMD experience a lower number of fractures, clinical observations reveal that many patients with VFs actually present with normal BMD scores. Recently, TBS measurement has been proven to be valuable in detecting VFs in patients with secondary osteoporosis [27]. We support the potential use of TBS as a complementary tool for assessing fracture risk, particularly in cases wherein BMD scores are normal. This may be attributed to degenerative changes in the lumbar spine, commonly seen in older patient populations which can artificially increase BMD values. Disease-induced alterations in trabecular structure could further contribute to an elevated risk of fractures. 

In our study, we observed that 3% of patients progressed to osteoporosis and 6% to osteopenia during the extended follow-up period, which contrasts with findings from other studies with shorter duration of follow-up [28]. The duration of TSH suppression duration varied considerably in this study, ranging from 15 to 85 months. Moreover, the transition rates from normal bone mineral density (BMD) and mild osteopenia to osteoporosis were reported as 0.6% and 3.2%, respectively, in this particular study.

The TBS exhibited a significant correlation with all analyzed BMD sites, particularly with L-BMD across all studies. However, when patients with osteopenia, osteoporosis, and normal BMD were categorized based on their TBS scores, significant discrepancies were observed. Patients with lower BMD tended to have a higher proportion of TBSs falling within the totally degraded and partially degraded ranges. In our study, correlations between BMI and bone parameters were identified (L-BMD r = 0.32, *p* = 0.0001 and TBS r = −0.41, *p* = 0.00009). Thus, an increase in adiposity around the region of interest may lead to a decrease in the signal-to-noise ratio, favoring lower TBS values. It is worth noting that the BMI of our non-obese patients fell well within the recommended working range for TBS (15–37 kg/m^2^) [29]. 

Our study has several limitations that warrant consideration. Firstly, it was conducted at a single site, which may limit the generalization of our findings. Additionally, the retrospective design of this study could potentially introduce bias when establishing the relationship between TSH suppression and its effects on bone. Although TSH levels were periodically assessed to ensure that they remained within the target range, fluctuations in TSH levels over time were not accounted for. Furthermore, data on factors such as the familiar history of fractures and falls were not collected, which could have provided valuable insights into fracture risk. Despite the identification of 30 fractures, the relatively small sample size may have limited our ability to detect statistically significant associations. However, this study also has strengths, including consistent follow-up by the same physicians and expert radiological analysis of vertebral fractures. To the best of our knowledge, this is the first study to compare TBS and DXA over a long follow-up period in postmenopausal DTC patients with maintained TSH suppression levels.

## 5. Conclusions

Our study highlights that, in postmenopausal women with DTC undergoing TSH-suppressive therapy, BMD experiences only minimal decline over a 12-year follow-up period, whereas the TBS sees a significant decrease. This suggests a potential association between TSH suppression and compromised bone quality. Interestingly, patients with VFs exhibit similar clinical and hormonal profiles to those without fractures, but demonstrate lower levels of physical activity and diminished TBSs. The correlation between TB scores with degraded microarchitecture and osteopenia underscores the importance of closely monitoring these patients for increased fracture risk.

## Figures and Tables

**Figure 1 diagnostics-14-00868-f001:**
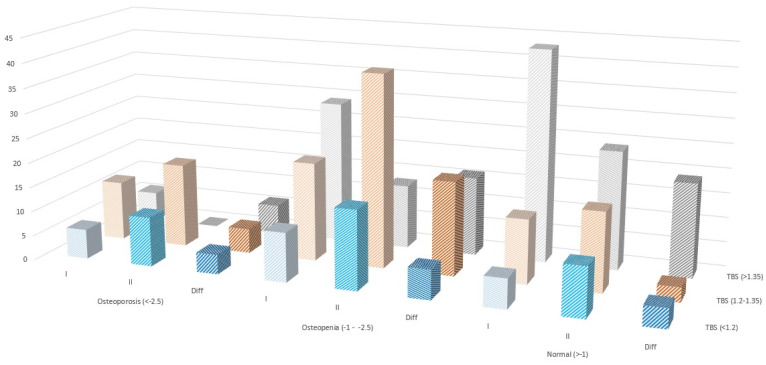
Results of the three-step classification approach for differentiated thyroid cancer patients with TSH suppression. The first step involves T-score classification: normal (>1 SD), osteopenia (−2.5 and ≤−1 SD), and osteoporosis (<−2.5 SD) with three subdivisions: I for initial visit, II for end visit, and DIF for difference. This step is based on TBS classification: normal (>1.35), partially degraded (1.2 and 1.35), and degraded (<1.2), represented linearly by a color. Patients in the osteopenia group with a partially degraded TBS were the largest subgroup showing an increase at the end of the study.

**Figure 2 diagnostics-14-00868-f002:**
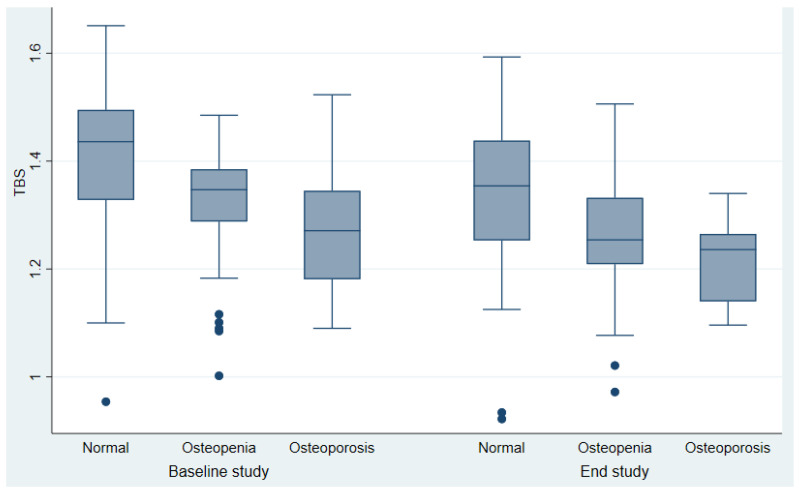
Long-term changes in TBS and BMD in 145 patients with differentiated thyroid cancer at baseline and the end of the study.

**Figure 3 diagnostics-14-00868-f003:**
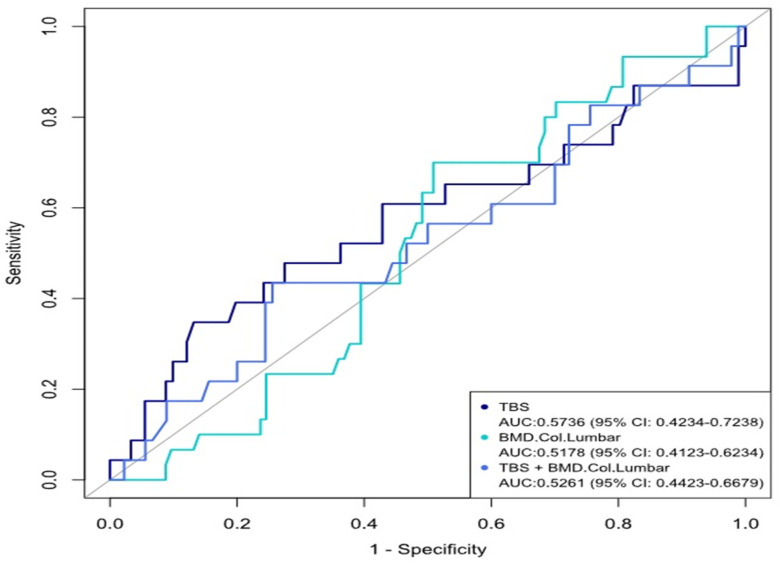
Area under the receiver operating curve determined by results of the logistic regression analysis, taking TBS and lumbar BMD as input parameters. Discriminative value of BMD alone and TBS alone or in combination with BMD in our cohort.

**Table 1 diagnostics-14-00868-t001:** Clinical and biochemical parameters of differentiated thyroid cancer patients at baseline and end of study.

Studied Parameters	Baseline Study (*n* = 145)	End of Study (*n* = 145)	*p* Value
Clinical and hormonal data			
Age (years)	51.48 ± 1.9	63.96 ± 10.65	<0.001
Weight (kg)	67.3 ± 11.8	70.28 ± 13.3	0.054
BMI (kg/m^2^)	27.27 ± 0.6	28.45 ± 5.3	<0.001
Menopause (years)	61 (57.9%)	131 (90.3%)	<0.001
Smoking yes/no	25 (17.2%)	17 (11.7%)	0.18
Alcohol ingestion	1 (0.69%)	0	0.32
LT4 (μg/kg)	2.29 ± 0.6	1.70 ± 0.4	0.0417
Serum TSH (µU/mL)	0.23 ± 0.4	0.89 ± 0.1	<0.001
Serum Free T4 (ng/dL)	1.64 ± 0.4	1.60 ± 0.3	0.9464
Duration years (range)	-	12.23 ± 5.9	
Radioactive iodine doses mCi (range)	-	209.17 ± 119.86	
Bone markers			
Serum PTH (pg/mL)	31.03 ± 12.1	45.65 ± 16.2	<0.001
Serum osteocalcin (ng/mL)	6.93 ± 3.5 (4.53–9.86)	19.56 (15.68–23.46)	<0.001
β-CTX (ng/mL)	-	0.30 (0.19–0.47)	
BAP (U/L)	10.90 ± 7.42	22.50 ± 4.9	<0.001
Serum 25OHD (ng/mL)	26.43 ± 10.1	22.65 ± 12.37	0.013
Densitometric parameters			
L-BMD (g/m^2^)	0.91 ± 0.16	0.89 ± 0.13	0.16
FN-BMD (g/cm^2^)	0.74 ± 0.14	0.70 ± 0.11	0.047
TH BMD (g/cm^2^)	0.91 ± 0.16	0.86 ± 0.13	0.42
UD-R BMD (g/cm^2^)	0.42 ± 0.06	0.40 ± 0.06	0.35
1/3 RD-BMD (g/cm^2^)	0.62 ± 0.05	0.63 ± 0.08	0.97
TBS	1.35 ± 0.14	1.26 ± 0.13	0.002
Normal	53.79%	26.21%	
Partial degraded	30.35%	48.97%	
Degraded	15.86%	24.83%	

BMI = body mass index; LT4 = levothyroxine; TSH = thyrotrophin hormone; Free T4 = free thyroxine; PTH = Parathyroid hormone; B-CTX = beta-CrossLaps; BAP = Bone alkaline phosphatase; 25OHD = 25 hydroxivitamin D; BMD = bone mineral density (L = lumbar; FN = femoral neck; TH = total hip; UD-R = ultra distal radius; 1/3 RD = third distal radius); TBS = trabecular bone score.

**Table 2 diagnostics-14-00868-t002:** Classification of patients with differentiated thyroid cancer according to DXA and TBS.

DXA T-Scores	TBS Scores			Total
	≥1.350 (Normal)	1.200–1.350 (Partially Degraded)	≤1.200 (Totally Degraded)	
T score ≥ 1 SD (normal)	43 (2.6)	13 (8.9)	6 (4.1)	62 (42.7)
T score < −1.0 and >−2.5 SD (osteopenia)	29 (20)	20 (13.7)	10 (6.8)	59 (40.6)
T score- ≤ 2.5 SD (osteoporosis)	6 (4.1)	12 (8.3)	6 (4.1)	24 (16.5)
Total	78 (53.7)	45 (31)	22 (15.1)	145 (100)

Number of patients (%). The change was statistically significant. Pearson Chi-square = 0.0057. T Scores were obtained by DXA (dual X absorptiometry). TBS = Trabecular bone score.

**Table 3 diagnostics-14-00868-t003:** Classification according to DXA and TBS at the end of the study after 12 years of follow-up in patients with differentiated thyroid cancer.

DXA T-Scores	TBS Scores			Total
	≥1.350 (Normal)	1.200–1.350 (Partially Degraded)	≤1.200 (Totally Degraded)	
T score ≥ 1 SD (normal)	25 (17.2)	16 (11.0)	9 (6.2) 10	50 (34.4%)
T score < −1.0 and >−2.5 SD (osteopenia)	13 (8.9)	38 (26.2)	17 (11.7)	68 (46.8)
T score- ≤ 2.5) SD osteoporosis	0 (0)	17 (11.7)	10 (6.9) 0	27 (18.6)
Total	38 (26.2)	71 (48.9)	36 (24.8)	145 (100)

Number of patients (%). Chi-Square = < 0.0001. The change rate was statistically significant: Pearson chi-square = 0.000. DXA = dual X absorptiometry. TBS = Trabecular bone score.

**Table 4 diagnostics-14-00868-t004:** Differences in overall BMD and TBS outcome at the end of the study in differentiated thyroid cancer patients.

Methods	BMD < −2.5 SD Osteoporosis			BMD −1–−2.5 SD Osteopenia			BMD >−1 SD Normal		
	Initial Study	End Study	Difference	Initial Study	End Study	Difference	Initial Study	End Study	Difference
TBS (<1.2) totally degraded	6	10	−4	10	16	−6	6	10	−4
TBS (1.2–1.35) partially degraded	12	17	−5	20	39	−19	13	16	−3
TBS (>1.35) normal	6	0	6	29	13	16	43	24	19

Number of patients. The change rate was statistically significant: Pearson Chi-square = 0.042. T Scores were obtained by DXA (dual X absorptiometry). TBS = Trabecular bone score.

**Table 5 diagnostics-14-00868-t005:** Comparison of patients with differentiated thyroid cancer with and without vertebral fractures at the end of the study.

	Without Vertebral Fractures	With Vertebral Fractures	*p*
Clinical and hormonal data			
*n*	115	30	-
Age (years)	52.0 ± 11.5	50.3 ± 13.5	0.50
BMI	26.6 (24.5–30.3)	25.8 (22.8–28.7)	0.32
Menarche (years)	12.95	12.96	0.97
Histology			0.40
Papillary	89 (85%)	29 (96.6%)	
Follicular	13 (11%)	1 (3.3%)	
Others *	4 (3.4%)	0 (0%)	
TNM initial stage			0.42
I	81 (71.8%)	18 (64.2%)	
II	15 (13.2%)	4 (14.2%)	
III	13 (11.5%)	6 (21.4%)	
IV	4 (3.54)	0 (0%)	
Osteoporosis	13/102	4/26	0.76
Exercise (walking minutes)	55.00 (30–60.00)	30.00 (30–45.00)	0.043
Milk ingestions (mg/day)	500 (477.50–750)	500 (250–750)	0.29
Smoking yes/no	19/96	6/24	0.65
Alcohol ingestion yes/no	1/114	0/30	0.61
Radioactive iodine doses: mCi (range)	150.00 (126–250.000)	150.00 (100–150.00)	0.042
Serum Biochemical parameters			
TSH (µU/mL)	0.03 (0.03–0.15)	0.10 (0.03–0.40)	0.20
T4L (ng/mL)	1.64 (0.42)	1.63 (0.45)	0.87
Tiroglobulin (ng/mL)	0.20 (0.00–1.00)	0.30 (0.00–1.20)	0.67
Calcium (mg/dL)	9.26 (0.57)	9.24 (0.50)	0.89
Phosphorus (mg/dL)	3.40 (3.0–3.90)	3.60 (3.3–4.0)	0.15
Creatinine (mg/mL)	0.71 (0.53–0.83)	0.75 (0.66–0.86)	0.50
PTH (pg/mL)	32.4 (0.63–0.83)	28.4 (24.5–40.6)	0.45
25OHD (ng/mL)	26.08 (17.0–33.4)	28.2 (22.2–33.6)	0.37
BAO (U/L)	10.77 (7.42–13.70)	11.30 (6.39–13.3)	1.00
Osteocalcin (ng/mL)	7.08 (4.66–9.89)	6.20 (3.74–9.2)	0.30
Densitometric parameters			
L-BMD	0.91 ± 0.17	0.88 ± 0.12	0.27
FN-BMD	0.75 ± 0.14	0.72 ± 0.12	0.56
TH-BMD	0.85 ± 0.11	0.83 ± 0.12	0.63
UD-R-BMD	0.44 ± 0.04	0.38 ± 0.06	0.12
1/3RD-BMD	0.63 ± 0.5	0.65 ± 0.5	0.72
TBS	1.35 ± 0.14	1.22 ± 0.14	0.001
Normal	53.9%	53.3%	
Partial degraded	29.8%	33.3%	
Totally Degraded	16.5%	13.3%	

* Others include 3 patients with Hurthle cell cancer (a variant of follicular carcinoma) and one patient with an encapsulated follicular variant of papillary carcinoma (a variant of papillary carcinoma). BMI = body mass index (kg/m^2^); PTH = parathormone; 25OHD = 25 hydroxy-vitamin D; TBS = Trabecular bone score. L-BMD, FN-BMD, TH-BMD, UD-R-BMD, 1/3 D-BMD = Lumbar, Femoral Neck, Total Hip, Ultra Distal, Third Distal Bone Mineral Density.

**Table 6 diagnostics-14-00868-t006:** TBS and DXA scores in 30 differentiated thyroid cancer patients with vertebral fractures.

	TBS			
DXA	Partial Degraded	Totally Degraded	Normal	Total
−2.5–−1	9 (30%)	5 (16.7%)	3 (10%)	17 (56.7%)
<−2.5	3 (10%)	0	0	3 (10%)
>−1	0	2 (6.6%)	8 (26.7%)	10 (33.3%)
Total	12 (40%)	7 (23.3%)	11 (36.7%)	30 (100%)

TBS = Trabecular bone score; DXA = dual X-absorptiometry. Fisher’s Exact Test = 0.0009.

## Data Availability

The data of these reported results are available from the corresponding author upon request.

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
