# Peer review of "Comparison of Bone Mineral Density and Trabecular Bone Score in Patients with and without Vertebral Fractures and Differentiated Thyroid Cancer with Long-Term Serum Thyrotrophin-Suppressed Therapy"

_diagnostics, 2024, doi:10.3390/diagnostics14090868_

Round 1

Reviewer 1 Report

Comments and Suggestions for Authors

The study by Federico Hawkins Carranza et al investigates the association between the risk of bone fracture and THS suppression. The study is interesting, but difficult to read due to poor writing quality

Title section: I suggest replacing “thyroid cancer” with “differentiated thyroid cancer”. It should also be outlined in the introduction section what types of thyroid cancer were taken into account

Line 20, 21, 26 – TBS, BMD, DXA, BMI – please explain the abbreviations when they appear in the text for the first time

Line 28 – “partially degraded and degraded TBS” – change to “partially degraded and totally degraded TBS”

line 58 – “This study aimed to analyze the concordance and difference evolution in TBS” – this is unclear. English language proofreading is necessary throughout the whole text.

Line 67 – “There were no included” - proofreading is necessary

Table 1 – no footnote

Table 2 and 3 – please put into the footer explanation of abbreviations used in the table (TBS/DXA etc)
All tables should have footnotes with the abbreviations  explained

Table 5 – “Menarchia”, “ingestión” – there are multiple mistakes throughout the manuscript

Table 5 – types of cancer – what cancers are meant by “other” cancers?

Line 263 and 264 – VFS and VFs – please be consistent with abbreviations

Comments on the Quality of English Language

There are multiple mistakes throughout the manuscript. Proofreading is necessary

Author Response

Reviewer 1:

The study by Federico Hawkins Carranza et al. investigates the association between the risk of bone fracture and TSH suppression. The study is interesting, but difficult to read due to poor writing quality.

 We thank you very much Reviewer 1,  for his comments. Following his suggestions we have carefully reviewed our manuscript to increase writing quality.

1.-Title section: I suggest replacing “thyroid cancer” with “differentiated thyroid cancer”. It should also be outlined in the introduction section what types of thyroid cancer were taken into account  

This has been done in the title(Page 1) and in the footnote of Table 5 to clarify this aspect.

2.-Line 20, 21, 26 – TBS, BMD, DXA, BMI – please explain the abbreviations when they appear in the text for the first time

This has been introduced in the text and also in the footnotes of all the tables.

3.-Line 28 – “partially degraded and degraded TBS” – change to “partially degraded and totally degraded TBS”

This has been done in tables 2,3,4,5 & 6 and in the text.

4.-line 58 – “This study aimed to analyze the concordance and difference evolution in TBS” – this is unclear. English language proofreading is necessary throughout the whole text.

We agree with Reviewer 1, and therefore we suppress the word concordance, to simplify our purpose in this study. Now is written “This study aimed to analyze changes in TBS, BMD and incidence of Vertebral Fractures (VFs) following a long-term follow-up of postmenopausal patients with DTC who underwent total thyroidectomy due to Differentiated Cancer  and were subjected to L-thyroxine (LT4) suppression therapy”-

5.-Line 67 – “There were no included” - proofreading is necessary

Thank you very much for the suggestion. In Section 2, 2.1 Study population, now we have written: “Patients were eligible for inclusion if they reached one-year post-menopause, and had maintained TSH suppression (<0.5 µU/mL) during 80% of their visits to our site. Informed consent was obtained from all patients. Those who had taken medications affecting bone metabolism (calcium or vitamin D, antiresorptive, hormone replacement therapy) or had other conditions potentially impacting bone metabolism (hyper and hypoparathyroidism, hyper and hypothyroidism, rheumatoid arthritis, asthma, Paget disease) were excluded from the study, as were those with abnormal kidney or liver function.

6.-Table 1 – no footnote.

We apologize, and now it is included in all the Tables.

7.-Table 2 and 3 – please put into the footer explanation of abbreviations used in the table (TBS/DXA etc)
All tables should have footnotes with the abbreviations explained.

This has been done in all the tables of the manuscript.

8.-Table 5 – “Menarchia”, “ingestión” – there are multiple mistakes throughout the manuscript

We apologize. We have corrected these writing errors in the text, including the Table 5                                                                                                                                      

9.- Table 5 – types of cancer – what cancers are meant by “other” cancers?

Others include 3 patients with benign Hurthle cell cancer (a variant of follicular carcinoma) and one patient with an Encapsulated follicular variant of papillary carcinoma (a variant of papillary carcinoma). (Haugen B.R.et al., Thyroid 2016)

This has been detailed in the footnote of Table 5.

 10.-Line 263 and 264 – VFS and VFs – please be consistent with abbreviations

We have corrected this mistake.

 11.-The manuscript would benefit from significant grammatical and typographical review. The quantity of grammatical and typographic errors does.

We apologize for the typographic and grammatical errors that could significantly impact the comprehension of the manuscript. We have asked for the revision of a professional Medical writer (Tradinfer S.L.)  to our manuscript.  With his help, we feel our manuscript has been improved in the English Language.

Reviewer 2 Report

Comments and Suggestions for Authors

This manuscript aimed to analyze the concordance and difference evolution in TBS, BMD, and Vertebral Fractures (VFs) after a long-term follow-up of postmenopausal DTC patients with total thyroidectomy due to differentiated thyroid cancer and Lthyroxine (LT4) suppressive therapy.It is a topic of interest to the researchers in the related areas. The paper needs very significant improvement before acceptance for publication. My detailed comments are as follows:

1.It is suggested that the author add the observation indicators of patients before thyroidectomy in the study, and exclude whether there is statistical significance in the difference of baseline data (gender, blood pressure, blood sugar) among the groups, so as to further clarify whether the above indicators have an impact on the experimental results

2.It is suggested that the author to improve the normal range of serum calcium, phosphate, creatinine, free T4, immunoglobulin, thyroglobulin, and other indicators

3.In the introduction part,It is recommended to discuss the mechanism of TSH for bone loss

4.In Tables 1 and 5, it is recommended to keep the same number of decimal places in the data for the same indicator and correct the punctuation in the article.

5.The number of Osteopenia in line 141 of the article is 69, while in Table 3 it is 68. It is recommended to unify the two and clarify the writing accuracy of 2OHD and T4L in Table 5.

6.the English language needs to be further revised.

7.It is advisable to indicate the meaning of P<0.001 under the table.

Comments on the Quality of English Language

the English language needs to be further revised.

Author Response

Reviewer #2:

This manuscript aimed to analyze the concordance and difference evolution in TBS, BMD, and Vertebral Fractures (VFs) after a long-term follow-up of postmenopausal DTC patients with total thyroidectomy due to differentiated thyroid cancer and L-thyroxine (LT4) suppressive therapy. It is a topic of interest to researchers in the related areas. The paper needs very significant improvement before acceptance for publication. My detailed comments are as follows.

We thank you very much Reviewer 2,  for his comments. We have introduced all the changes suggested by Reviewer2, for the improvement of the manuscript. The research purpose was to analyze our long-term experience with these two approaches to the long-term follow-up of bone health in thyroid patients in, our Thyroid Cancer Unit.

1.-It is suggested that the author add the observation indicators of patients before thyroidectomy in the study, and exclude whether there is statistical significance in the difference of baseline data (gender, blood pressure, blood sugar) among the groups, to further clarify whether the above indicators have an impact on the experimental results.

In Table 1. baseline clinical and biochemical data before thyroid surgery were compared with the end study. Not all clinical parameters were included because they were not the purpose of the study, but as patients were followed in our Thyroid Unit, we can confirm that there was no hypertension, or diabetes in patients during the follow-up.  Regarding gender, only postmenopausal women were included in the study, and a subanalysis of the incidence of vertebral fractures was performed.

2.-It is suggested that the author to improve the normal range of serum calcium, phosphate, creatinine, free T4, immunoglobulin, thyroglobulin, and other indicators

Now in the material and methods sections, we have written the normal range of the parameters. We have not included immunoglobulin, as they were not our objectives.

3.-In the introduction part, It is recommended to discuss the mechanism of TSH for bone loss

Following Referee 2 suggestions, in the introduction section, there is a new paragraph regarding the mechanism by which TSH can induce bone loss:

The mechanism underpinning bone loss remains poorly understood, but it is posited that suppressed TSH levels could impede bone formation while promoting osteoclast differentiation leading to dysregulated bone turnover and microarchitectural abnormalities(4).

4.-In Tables 1 and 5, it is recommended to keep the same number of decimal places in the data for the same indicator and correct the punctuation in the article.

Thank you very much. Now this has been corrected.

5.- The number of Osteopenia in line 141 of the article is 69, while in Table 3 it is 68. It is recommended to unify the two and clarify the writing accuracy of 2OHD and T4L in Table 5.

We apologize for this mistake. It corresponds to 68 in line 141.

6.-The English language needs to be further revised.

We apologize for the style and grammatical errors in the submitted manuscript. A native English translator has reviewed the text

7.-It is advisable to indicate the meaning of P<0.001 under the table.

This is now detailed in the statistical section.

Round 2

Reviewer 1 Report

Comments and Suggestions for Authors

1.-Title section: I suggest replacing “thyroid cancer” with “differentiated thyroid cancer”. It should also be outlined in the introduction section what types of thyroid cancer were taken into account

This has been done in the title (Page 1) and in the footnote of Table 5 to clarify this aspect.

I previously suggested changing the "Thyroid Cancer" in title to "Differentiated Thyroid Cancer"

2.-Line 20, 21, 26 – TBS, BMD, DXA, BMI – please explain the abbreviations when they appear in the text for the first time

This has been introduced in the text and also in the footnotes of all the tables.

I can't see any changes here. There are some changes in the file I downloaded from the website ("diagnostics-2947915-peer-review-v2"), but I can't see any of my suggestions from commentary 2 applied to the manuscript.

8.-Table 5 – “Menarchia”, “ingestión” – there are multiple mistakes throughout the manuscript

We apologize. We have corrected these writing errors in the text, including the Table 5

"Menarchia" still remains uncorrected. I think this is a Spanish word for "menarche"

Comments on the Quality of English Language

The quality of English has improved, however some minor corrections are still necessary

Author Response

Reviewer 1:

Comments and Suggestions for Authors:

1.-Title section: I suggest replacing “thyroid cancer” with “differentiated thyroid cancer”. It should also be outlined in the introduction section what types of thyroid cancer were taken into account.
I previously suggested changing the "Thyroid Cancer" the title to "Differentiated Thyroid Cancer"

This has been done in the title (Page 1) and in the footnote of Table 5 to clarify, as was required, and by an elfin mistake not introduced. In the introduction section (line 55)we have introduced “due to DTC (Papilar and Follicular types)”

2.-Line 20, 21, 26 – TBS, BMD, DXA, BMI – please explain the abbreviations when they appear in the text for the first time

This has been introduced in the text and also in the footnotes of all the tables. By a typewriting mistake, not introduced. Now are written in the footnotes of all the tables.

I can't see any changes here. There are some changes in the file I downloaded from the website ("diagnostics-2947915-peer-review-v2"), but I can't see any of my suggestions from commentary 2 applied to the manuscript.

8.-Table 5 – “Menarchia”, “ingestión” – there are multiple mistakes throughout the manuscript."Menarchia" still remains uncorrected. I think this is a Spanish word for "menarche"

We apologize. Now we have written menarche in Table 5.  The inadequate accent in ingestion corresponds to the Spanish concealer

We have carefully reviewed again our manuscript to fulfill all the queries and recommendations you have sent. We are very sorry and apologize, because for unknown reasons some of your suggestions that were done, were not engraved in the file.

Sincerely yours,

Federico Hawkins Carranza